# Early empathy development: Concern and comforting in 9- and 18-month-old infants from Uganda and the UK

Carlo Vreden[1,2*‡], Joanna C. Buryn-Weitzel[3‡], Santa Atim[4], Ed Donnellan[3,5], Maggie Hoffman[3,6], Eve Holden[1,3], Michael Jurua[4], Charlotte V. Knapper[3], Nicole J. Lahiff[3,7], Sophie Marshall[3], Josephine Paricia[4], Bahar Tuncgenc[8], Florence Tusiime[4], Claudia Wilke[3], Katie E. Slocombe[3], Zanna Clay[1*]

**1** Department of Psychology, Durham University, Durham, United Kingdom, **2** DIPF | Leibniz Institute for Research and Information in Education, Frankfurt, Germany, **3** Department of Psychology, University of York, York, United Kingdom, **4** Budongo Conservation Field Station, Masindi, Uganda, **5** Department of Psychology, University of Warwick, Coventry, United Kingdom, **6** School of Human Evolution and Social Change and Institute of Human Origins, Arizona State University, Tempe, Arizona, United States of America, **7** Department of Evolutionary Anthropology, University of Zurich, Zürich, Switzerland, **8** Department of Psychology, Nottingham Trent University, Nottingham, United Kingdom

‡ These authors are joint first authors on this work.
* zanna.e.clay@durham.ac.uk (ZC); c.vreden@dipf.de (CV)

## Abstract

Empathy, the capacity to share and understand others' emotional states, is important for navigating our social lives, yet its development in infancy remains poorly understood. Moreover, most research on empathy development has been biased towards Western populations, which are not representative of global diversity. Using a cross-cultural longitudinal design, this study investigated how infants at 9 and 18 months sampled from Uganda (N = 44, 24 female) and the UK (N = 49, 24 female) develop empathic concern and comforting. Infants watched an adult (mother or experimenter) pretend to injure themselves and the infants' concerned facial expression and comforting behaviours towards the injured individual were recorded. By 9 months of age, infants at both sites exhibited evidence of facial expressions of concern and onset of comforting behaviour. The likelihood of comforting at both sites increased by 18 months. Results were overall similar at both sites, but we found some site level variation in tendencies to offer comfort spontaneously, with infants in Uganda being more likely to comfort after an explicit cue of need than infants in the UK. Overall, results highlight early onset of empathy development in infancy, with similar developmental trajectories in two sites, despite differences in socio-cultural environment.

**Data availability statement:** Data can be accessed at https://osf.io/2sek5/?view_only=82414d1b18eb42adbbe17b09c473dc35

**Funding:** This research was funded by the European Research Council Consolidator Grant awarded to KS (724608) and the European Research Council Starting Grant awarded to ZC (802979). The funders had no role in study design, data collection and analysis, decision to publish, or preparation of the manuscript.

**Competing interests:** The authors have declared that no competing interests exist.

## Introduction

Empathy, broadly defined as the sharing and understanding of others' emotional states and thoughts, is a core component of human socio-emotional functioning [1,2]. Despite the important role empathy plays in shaping social interactions and responses to others' emotional needs, understanding of its early development, especially during the first year of life, remains limited. Additionally, a persistent bias in the existing literature towards industrialised Western samples means we know relatively little about the extent to which empathy development is shaped by socio-cultural factors.

Empathy is broadly considered a multi-dimensional process involving both affective and cognitive components [3]. These include the *affective* sharing or experiencing of others' emotional states, through evolutionarily ancient processes such as emotional contagion and mimicry [2]. The *cognitive* components of empathy relate to understanding others' states, via perspective-taking and self-other differentiation. There remains active debate as to when and how such processes emerge and the extent to which the cognitive and affective components are dissociable, or whether they represent an integrated system [4–6].

Studies with infants during the first year of life primarily focus on two main behavioural markers of empathy: facial expressions of concern and comforting behaviour. Although mapping external behaviours onto underlying mechanisms is challenging, it seems plausible that facial expressions of concern may reflect emotional contagion or a deeper understanding of others' emotional needs. In contrast, comforting seems to be a clearer indicator of understanding emotional needs in others and has been commonly used as a marker of empathy in both children and adults. Facial expressions of concern following simulated distress by an adult have been observed in 8-month-olds from Israel and the US [7,8]. Although promising, a lack of control condition in these studies makes it difficult to rule out whether such expressions might reflect concern about lack of attention and normal responsivity in the adult or the infant's own distress rather than concern for the adult. In a more recent study [9], the inclusion of a control condition of a neutral adult not paying attention to the child showed that even by 3 months, infants showed more signs of facial concern in response to a distressed adult than a neutral adult (albeit weakly and briefly at this age). Facial concern was found to increase with age but with steepness gradually decreasing over time [7–9]. Supporting the integrated view of empathy, the early trajectory of facial expressions of concern appear to predict other aspects of empathic development, including comforting behaviour [9] but also more broadly to social competence [10].

In contrast to facial expressions of concern, comforting behaviours (such as hugging the distressed individual) appear later in development. For instance, comforting has not been detected at 3 and 6 months in Israeli infants [9]. It then appears to become more likely over the second year of life (e.g., from 14% of infants at 12 months to 41% of infants at 18 months, [9]). However, another study [11] found no evidence of comforting behaviours in 18- and 24-month-old Canadian infants who witnessed a 10-second-long distress simulation by an experimenter, though this

could have been due to the short duration of the simulation. Overall, the evidence suggests that across the second year of life, comforting behaviours are observable but can still be rare and sensitive to context, whereby children may need sufficient time to both process and respond to observed distress. It is also possible that such behaviours could potentially represent self-comforting rather than other-oriented responding [7] and control conditions can be important in ruling out comforting behaviour being driven by a desire to simply elicit a response from an unresponsive adult [9,11].

Another key issue hampering progress in our understanding of empathy development is a persistent sampling bias towards a limited number of cultural settings, particularly high-income, educated families in industrialised settings, particularly in Western Europe and North America. This sampling bias, which is persistent across psychology, including developmental research [12,13], ignores the rich and important socio-cultural variability that shapes the social nature of our species. Only by addressing it and including more culturally diverse samples can the foundations of complex capacities, such as empathy, be fully understood.

The importance of including more diverse samples becomes apparent when considering the handful of studies on empathy development that have already been conducted in more than one sociocultural setting. These have found evidence for cross-cultural variation in empathy, albeit mostly beyond infancy: For example, 6- to 7-year-old US-American children matched the emotional state of characters in affective stories more closely than did children from Greece [14]. 7- to 12-year-olds from Brazil scored higher on an empathy questionnaire than 6- to 13-year-olds from the US [15], suggestive of possible cross-cultural variation in the expression of empathy during middle and late childhood. When might such variation start to emerge? In a study on Malaysian, Indonesian, German, and Israeli 5-year-olds [16], all showed similar intensity of empathic concern when seeing an adult in distress over a broken toy. However, Malaysian and Indonesian children exhibited more self-distress and fewer empathic behaviours than German and Israeli children, suggesting that some cross-cultural differences in empathy may already be present in earlier childhood. A similar study has been conducted with infants, in which 19-month-olds from urban Germany and urban India witnessed an adult being upset over a broken toy [17]. Children from both contexts displayed self-distress and other-oriented empathic behaviours, such as comforting or engaging another adult's help, without significant differences between the two sites, despite variation in maternal prosocial and socialisation goals. It should be noted, though, that this study [17] measured empathic concern and comforting behaviours in one combined score. It is therefore unclear whether early cross-cultural differences might exist for one marker of empathy but not the other. Taken together, these previous cross-cultural studies highlight the potential for important variation in the development and expression of empathy in early childhood, yet many questions remain. Importantly, only a single cross-cultural study [17] has examined empathy in infancy, so the age at which empathy development starts to show sensitivity to the cultural environment needs considerably more investigation. Most previously studied samples were (as far as that information is available) from urban, industrialised, middle-class settings, providing some homogeneity. To our knowledge, no such study has been conducted in a rural subsistence context, leaving a significant gap in our knowledge of variation in the development of empathy. Given that empathy is considered a key facet of socio-emotional development, an increased effort to understand it in more diverse contexts is needed if we want to fully understand its developmental origins and trajectory.

There are several reasons why cultural variation may be seen in early empathic behaviours. Variation in socialisation goals and practises of parents and other caregivers may affect the development of empathic behaviours in a number of ways. It has been proposed that observed differences in socialisation practices arise from differences in the cultural models of the broader community [18,19]. For instance, individuals from relational sociocultural contexts have been found to put more value on interpersonal relatedness, responsiveness, and obedience than individuals from autonomous communities, who tend to put more emphasis on individuality and independence (e.g., [17,20]). It is plausible that infants of caregivers with more relational socialisation goals attend more to the needs of others. It is also possible that variation in socialisation of emotional expression and regulation could affect the development of empathetic behaviours. Emotion regulation is considered important for expressing other-oriented empathic responses [21,22], and cross-cultural variation

in the socialisation of emotion regulation has been found. For instance, comparing Germany and Cameroon, German mothers prioritised having emotionally expressive children, whereas Cameroonian Nso mothers placed more value on their children being calm and emotionally inexpressive [23].

Given the need for more diverse infancy research, the aim of the present study was to systematically investigate the early development of empathy in two distinct socio-cultural settings – participants sampled from Uganda and the UK. These samples varied in potentially relevant ways, as outlined below, however it was beyond the scope of this study to examine whether individual variation in these factors was predictive of infant empathetic behaviour: this should be a focus of future research. The Ugandan site consisted of multiple villages set on the edge of a large rainforest. Most people in the area rely on subsistence farming, as opportunities for and engagement in wage labour is low. Families often live in multi-generational households and distributed caregiving, including the presence of child caregivers, is common [24]. This site in Uganda poses an interesting context to study socio-emotional development, as ethnographic research in Uganda and other rural African contexts has documented the concealment of emotions and generally low levels of affect and emotional engagement in mother-infant relationships [25–27]. Additionally, Ugandan mothers from this research site have been shown to value relational over autonomous goals more than mothers from the UK site [24] – two cultural values which have been shown to be linked to differences in emotion socialisation [23]. Socio-cultural differences between our sites in Uganda and the UK with respect to an infant's learning environment, caregiving arrangements, and maternal socialisation goals and strategies, allow for an investigation into the previously established trajectories for empathy development outside of and within traditionally studied populations.

Building on previous work (e.g., [7,9,11]), in this study, we conducted a comforting experiment when infants were 9 and 18 months old. In the experiment the infants observed an adult model (mother or adult experimenter) lightly injuring themselves. In the experimental condition, the model expressed distress after the pretend injury and did not respond to communication attempts by the child. Following Dunfield et al.'s protocols [11], in the control condition, the model did not express any signs of distress following the injury but remained emotionally neutral and unresponsive to the child. Although other previous studies have implemented other types of control conditions (e.g., [7,9]), we opted to include this control condition for two reasons: first, to test whether the child acted in response to the adult's distress signals that express a need for comforting and not just the injury itself (as 18-month-olds have been shown to respond empathically after an injury even in the absence of distressed signalling [28]); and second, to rule out that comforting during the experimental condition was not just an attempt to elicit a social response from an unresponsive adult. To measure behavioural markers of empathy, we coded the frequency and duration of two behavioural components in the infants: facial expressions of concern and comforting behaviours. Many previous studies (e.g., [7,9]) have also included 'hypothesis testing', i.e., inquiry behaviours thought to represent attempts to understand the distress. A key behavioural marker of hypothesis testing is child gaze between the adult's hurt body part and face or another adult. As our study included the distressed adult gaze alternating between their hurt body part and the child, this may have encouraged gaze alternation in the child, making interpretation of this behavioural marker very difficult. Therefore, we only present the data relevant to hypothesis testing in the supplementary materials, as we are not confident that they represent a valid measure of hypothesis testing within our paradigm.

We predicted that by 9 months of age, infants across both sites would display concerned facial expressions in response to an adult in distress but would not yet show comforting behaviour [7,9]. We expected that both concerned facial expressions and comforting would be present at 18 months, and that a greater proportion of children would show these behaviours at 18 months compared to 9 months [7,9,29]. Since findings on the predictiveness of early empathic behaviours from one timepoint to the next are mixed (e.g., [7,9]) we took an explorative approach to assessing if individual variation in concerned facial expression at 9 months would predict comforting at 18 months. Moreover, as comforting, but not concerned facial affect, has previously been found to be socially biased towards familiar individuals [29], we expected to see more comforting behaviour (but not concerned facial expressions) towards the mother than the experimenter in both socio-cultural contexts.

Regarding potential cultural variation, we took an exploratory approach, given limited previous research in rural Uganda and at this site in particular. We identified the following possible drivers for population-level differences in early empathetic behaviour: Sociocultural contexts emphasising social connectedness (i.e., relational models, as documented at this site in Uganda; [24]) may foster earlier empathic behaviours but may also hinder the development of empathic behaviours if repression of negative emotions is valued and thus fewer opportunities to respond to those in distress are provided [23]. Additionally, the higher value placed on emotional expressivity in autonomous contexts, such as the UK, may encourage more attention to and thus an increased tendency to respond empathically to others' emotional states. These competing influences could plausibly make infants either more or less likely to show facial expressions of concern and comforting.

## Methods

### Participants

Participants were all part of a larger longitudinal study on the development of social cognition and communication from 3–24 months. The sample size in each population was therefore based on the sample size of this larger longitudinal study (which was informed by a power analysis), and by the dropout rate from the longitudinal study. Therefore no a priori power analysis was performed for this specific study. Forty-four (24 female, 20 male) mother-infant-dyads from Nyabyeya parish, Masindi district, Uganda (hereafter Uganda), and 49 (24 female, 25 male) from in and around the city of York, United Kingdom (hereafter UK), recruited between 22/9/17 and 22/9/18, participated in this study, and provided a total of 464 valid trials. Refer to supplementary materials (S1 in S1 File) for how many valid trials were retained at each timepoint and testing site after exclusions due to experimental errors.

Of the Ugandan mothers for which ethnolinguistic information was available, 42% identified as Alur, 27% as Lugbara, and 31% as belonging to other ethnolinguistic groups (Banyuro, Kakwa, Kaliko, Kebu, Lendu, and Ma'di). UK mothers identified as 77% white British, 12% British (not further specified), 9% white (not further specified), and 2% mixed British. Most Ugandan mothers had some primary school education (64%), 19% had some secondary education, and 17% had no formal education. Most UK mothers (86%) had completed post-secondary qualifications (undergraduate or equivalent degrees, or higher), and 14% had completed secondary education. In Uganda, 28% of mothers had an additional income besides subsistence farming (e.g., selling crops for money or tailoring) but none were formally employed, while in the UK all mothers were in formal employment prior to maternity leave. At their infant's birth, the Ugandan mothers were 15–42 years old ($M_{age}$ = 26.42 years, $SD$ = 6.79) and the UK mothers were 25–40 ($M_{age}$ = 32.68 years, $SD$ = 3.66). Twenty-seven of the Ugandan and 24 of the UK infants had at least one older sibling. Mean parental Hollingshead SES in Uganda was $M$ = 8.35 ($SD$ = 6.65; range = 0–28) and in the UK $M$ = 53.30 ($SD$ = 10.28; range = 24.5–66).

The data for this study were collected during one 2-hour-long session when the infants were 9 months old (Uganda: $M_{age}$ = 8.62 months, $SD$ = 0.49; UK: $M_{age}$ = 8.50 months, $SD$ = 0.51) and two 2-hour-long sessions when the infants were 18 months old (Uganda: $M_{age}$ = 17.74 months, $SD$ = 0.45; UK: $M_{age}$ = 17.41 months, $SD$ = 0.52). These sessions included a variety of different tasks and measures on the infants' socio-cognitive development besides the comforting task reported here.

### Ethics

Approvals were obtained from the Department of Psychology Ethics Committees at the University of York and Durham University, the Ugandan Virus Research Institute Regional Ethics Committee, and the Ugandan National Council for Science and Technology (reference number: SS4545). Mothers gave overall consent for their and their infant's participation in the longitudinal study. Consent for the individual tasks, including comforting, was additionally obtained. UK mothers read the information sheet and gave written consent. To accommodate lower literacy levels among the Ugandan participants, a local research assistant fluent in both English and the mother's primary language read the information and consent forms out loud to the mother and the mother could choose to give a written signature or a thumbprint, as is standard in Uganda.

Additional information regarding the ethical, cultural, and scientific considerations specific to inclusivity in global research is included in the supplementary materials (S10 in S1 File).

### Design

The task was based on prior work [11], which was originally developed for families living in urban Canada. Although observing injuries is a common day-to-day event in any culture, we ensured cultural appropriateness of the task in Uganda and the UK by piloting the task and by co-producing the final version of the task with local Ugandan team members (SA, JP, FT, MJ) along with researchers from the University of York (MH, JBW, EH, ED, CW).

At 9 months old, infants only participated in the comforting task with their mothers. Practical constraints on the length of the testing visit at this age meant an additional experimenter condition was not feasible. At 18 months, infants completed the experiment twice, to examine the effects of model familiarity: once with a local experimenter, and once with their mother as models (conducted on different days). For both model types, infants took part in an experimental condition and a control condition, with breaks and other tasks (not reported in this study) in-between the conditions, and at least one full day between the mother and experimenter trials, to aid credibility of the distress simulation.

There were four possible scenarios for the experimental task, to avoid habituation. The scenarios were as follows: hitting a finger with a toy hammer (scenario A), pinching a finger in a clipboard (scenario B), hitting a foot on a piece of furniture (scenario C), and bumping a knee into a piece of furniture (scenario D). The scenario was identical for the control and experimental condition but varied between timepoint (9 and 18 months) and model identity (mother and experimenter). In Uganda, only scenarios A and B were used, as testing took place outside (as was culturally appropriate) and there were no pieces of furniture to credibly bump into. As we aimed for children to complete 3 trials (1 at 9 months; 2 at 18 months), it was necessary in Uganda to use one of the two scenarios twice. To aid comparability across sites, we thus ensured in the UK and Uganda that the scenario used for the mother at 9 months was then also used for the experimenter at 18 months and a new scenario was introduced for the mother at 18 months. The order of the conditions, the scenario, and the model doing the task first (i.e., mother or experimenter) were counterbalanced across participants.

### Materials

A yellow plastic toy hammer and a black clipboard were used in scenarios A and B. The task was filmed from two angles, one focusing on the infant and the other on the mother/experimenter. In Uganda, Panasonic HC-VX870 4K camcorders with external microphones (Sennheiser MKE 400 Shotgun Microphone) were used. In the UK, the task was filmed with Panasonic HC-VX870 4K and Panasonic HC-V HD camcorders.

### Procedure

Data were collected during home visits by teams of 2–3 researchers, which in Uganda included at least one local researcher. In both the UK and Uganda, researchers leading the visits spoke a mutually fluent language with the infant's mother. In Uganda, most testing took place outside, in the family compound. In the UK, all testing took place indoors, in the room that the mothers selected as most appropriate (the living room in most cases).

Researchers at both sites practiced their 'accidents' and subsequent reactions until they were fluent and realistic. Before the start of each trial, mothers were trained on how to act in the experiment by a researcher who gave a demonstration, asked the mother to practice (out of sight/earshot from infant), and gave her feedback. Trials where mothers broke character (e.g., laughing), or did not follow the cues, were excluded prior to video coding (see supplementary materials S1 in S1 File).

Each trial began with the infant sitting on the floor, either on their own or, if necessary, supported by an adult who was not otherwise part of the task. The presence of an additional adult was required in ca. 10% of trials to keep the infant in

view of the cameras before the adult in distress started their simulation. Once the trial had begun (i.e., the distressed adult had started to act), the additional adult moved so they were no longer in physical contact with the infant, were turned away from the infant and engaged in an unrelated activity so they were not available to comfort the hurt adult or interact with the infant. The model (i.e., mother or experimenter) started the trial sitting on the floor opposite the infant in scenarios A and B or standing next to a piece of furniture in scenarios C and D, facing the infant. The model first got the infant's attention by calling their name and then pretended to accidentally walk into a piece of furniture or to hurt their finger. In the experimental condition, while pretending to get hurt, the model gave a culturally appropriate exclamation of pain (e.g., "Ouch!" in UK or gasping or sucking in of air in Uganda). In the control condition (based on [11]), after the action, the model gave an exclamation of mild surprise (e.g., "Ooh!"). After the pretend accident had happened, in scenarios C and D, the model sat down on the floor in front of the infant. In scenarios A and B, the model remained sitting opposite the infant and hid the toy hammer or clipboard behind their back to avoid the infant being distracted by the item during the rest of the trial.

In the experimental condition, after the initial distress vocalisation, the model continued to vocalise as if in pain whilst rubbing and gazing at their hurt body part, then gaze alternated between this and the child's face (based on [11]). They then stopped these cues, looked at the child and said "Look, I hurt myself!" once after ca. 30 seconds. Gestural and verbal pain cues were accompanied by a pained facial expression. In the control condition at 9 months, the same gaze patterns were applied but the model did not make any noise, held a neutral expression throughout the trial, and did not rub their hurt body part. In the control condition at 18 months UK models pretended to read a piece of paper and Ugandan models looked at their nails. Due to low literacy rates in the communities, reading was not an appropriate distraction activity for the mothers in Uganda. Ugandan researchers advised that examining one's hands was a common activity which would divert visual attention in a similar way, so this adaptation was implemented. Trials in both conditions lasted ca. 40s from the model's injury until the end of the trial. See supplementary materials for the full experimental script (S2 in S1 File). Although the procedure of the current study was based on prior work with a 10s trial duration [11], we opted to extend the distress event in length, to give infants, especially younger ones with less developed motor skills, more opportunity to comfort the distressed model.

To standardise the timing of the distress cues given by the model in the experimental condition, all experimenters and UK mothers wore Bluetooth earpieces over which a recording was played that reminded them of what to do when. In Uganda, as this technology was not familiar to participating mothers, the researcher listened to the recording and then quietly instructed the mothers. Whenever possible this was done in a different language than the one the mother used when speaking to her child.

## Coding and data preparation

Infant behaviours were coded in ELAN [30], an open-source audio and video annotation tool accompanied by a bespoke MATLAB [31] script to identify timings and cue contingencies.

A second UK-based coder not otherwise involved in the study coded 20% of all trials for concerned facial expressions and comforting across both sample populations. Inter-observer reliability was good with Cohen's kappa at 0.88 for the occurrence of comforting (categorical yes/no occurrence measure) and an intraclass correlation coefficient of r = 0.83 for duration of concerned facial expressions (continuous measure in seconds). Because comforting rates were relatively low, we wanted to reduce potential type-I errors, therefore all cases of comforting identified by the primary coder (CV) were checked by another coder (JBW), with a kappa of 0.79. All cases of disagreement were reviewed by the two coders and if agreement could not be reached, they were adjudicated by a third coder. As interpretations of facial expressions may potentially vary cross-culturally [32], we discussed the coding scheme with Ugandan researchers (SA, JP, FT, MJ), and they agreed that the descriptions accurately described the facial expressions they associated with concern. A Ugandan coder (FT) then coded 20% of the Ugandan trials for instances of concerned facial expression. CV and FT's kappa value reached 0.73, indicating that the UK-based coding of Ugandan facial expressions had been done reliably.

Below, we overview the coded behaviours; the full coding scheme, based on an established coding scheme [33], which has been used extensively in empathy development research, is available in the supplementary materials (S3 Table in S1 File). Each behaviour was coded for both experimental and control conditions, from when the model pretended to get hurt until they stopped giving cues.

**Concerned facial expression.** Concerned facial expression was coded as frowning, raised or drawn together eyebrows, widened eyes, or tight lips [33]. Since both self-distress and empathic concern may produce very similar facial expressions in infants, facial expressions were only coded as concern if they did not co-occur with clear signs of self-distress from the infant (i.e., crying). Concern was not coded for periods where the participant's face was not visible (i.e., if at least one half of the face was clearly not visible from at least one camera angle for more than half a second). Concerned facial expression was coded with exact on- and offsets. We then extracted its proportional duration relative to the trial length, subtracting periods of the trial where the infant's face was not visible. We coded duration of concerned facial expression, rather than intensity as previous studies have often done (e.g., [7,9]), to account for potential differences in emotional expressivity between the two-cultural settings, so that potentially culturally less expressive infants would not be coded as less empathic.

**Comforting.** For each trial, we coded all instances of empathic behaviour in the form of comforting the distressed individual. Comforting, or attempts to comfort, included gentle contact actions oriented towards the model – e.g., infant pats, strokes or hugs the model, gives a nearby toy or other object to the model – and verbal responses, such as "Oh oh!" or "Ouch!" [33]. Participants received a binary score per trial for occurrence of comforting. Although previous ratings schemes have used 0–3 ratings of comforting intensity or urgency [33], these were not developed with culturally varied expressions of comforting in mind.

**Cue associated with comforting behaviour.** As comforting was rare at 9 months, we based our cues analysis only on comforting at 18 months. For infants who comforted at 18 months, we extracted which of the standardised distress cues immediately preceded the infant's comforting behaviour. Given that we were interested in infant responses occurring before or after explicit labelling of distress, we collapsed the cues into two categories: infants who comforted in the experimental condition before the model labelled their distress ("Look, I hurt myself!") and those who comforted afterwards.

## Statistical analyses

A total of 465 valid control and experimental trials across the two sites and timepoints were available for analysis. Whilst some infants had valid trials at 9 months with their mother and valid trials with both the mother and experimenter at 18 months, other infants did not have the full set of trials due to missing trials, or trials being excluded (see supplementary materials S1 in S1 File for details). All measures were extracted for each valid trial.

We first tested the effect of condition and if this was influenced by age or site to ensure our experimental condition had an effect on infant behaviour. Then, for the experimental condition only, we examined (i) the effects of age and site, (ii) the effect of familiarity with the distressed adult and site on the proportion of the trial where the infant displayed concerned facial expressions, and on the likelihood of the infant comforting; and (iii) whether showing a concerned facial expression at 9 months predicted comforting at 18 months (because comforting at 9 months was extremely rare, we did not expect to find a predictive relation between comforting at the two timepoints).

All analyses were conducted in R [34] using the package lme4 [35] for GLMMs. For each statistical model, we compared the full model, which included an interaction term, to a null model lacking the fixed effects and interaction terms of interest but retaining control fixed effects and random effects using a likelihood-ratio test [36]. If this was significant, we compared the full to a reduced model, where the interaction term was removed and constituent factors were entered as main effects, also using a likelihood-ratio test. The drop1 function was used to extract p-values for individual variables. See Table 1 below for model structures. See supplementary materials (S4 in S1 File) for model assumptions and stability

**Table 1. Overview of model structures for full model GLMMs and GLMs.**

| Research Question | Outcome | Fixed effects: Predictor variables | Fixed effects: Control variables | Random effects | Error structure and link function |
|---|---|---|---|---|---|
| Condition effect | Concerned facial expression | Condition*site, condition*age | Infant sex | Participant ID | Beta error distribution and logit link function |
| | Comforting | Condition*site, condition*age | Infant sex | Participant ID | Binomial error structure, logit link function |
| Age and site effects | Concerned facial expression | Age*site | Infant sex | Participant ID | Beta error distribution and logit link function |
| | Comforting | Age*site | Infant sex | Participant ID | Binomial error structure, logit link function |
| Familiarity and site effects | Concerned facial expression | Familiarity*site | Infant sex | Participant ID | Beta error distribution and logit link function |
| | Comforting | Familiarity*site | Infant sex | Participant ID | Binomial error structure, logit link function |
| Facial concern predicting later comforting | Comforting at 18 months | Concerned facial expression (9m)*site | Infant sex | None | Binomial error structure, logit link function |

assessment procedures. To test the robustness of our findings we also ran Bayesian equivalents of all generalised linear mixed models (see supplementary material S9 in S1 File for details).

We also examined the spontaneity of the comforting response and the influence of explicit distress cues at 18 months. As comforting behaviour remained rare, even at 18 months, we maximised our sample by including data from both mother and experimenter trials into one analysis. Since there was no case where an infant at 18 months comforted both their mother and the experimenter, pseudo-replication was not an issue. We used a chi-squared test to compare the distribution of infants who comforted in Uganda vs. the UK before the mother or experimenter explicitly labelled their distress vs. afterwards.

## Results

Table 2 shows descriptive statistics for concerned facial expression and comforting, by age, site, and identity of the distressed model, in the experimental and control conditions. See supplementary materials for a more detailed overview of specific comforting behaviours (S5 in S1 File), results from hypothesis testing (S6 in S1 File), and full model estimate

**Table 2. Descriptive statistics for concerned facial expressions and comforting, across conditions, sites, and timepoints.**

| | Condition | Uganda | | | UK | | |
|---|---|---|---|---|---|---|---|
| | | 9 months | 18 months | | 9 months | 18 months | |
| | | Mother | Mother | Experimenter | Mother | Mother | Experimenter |
| Concerned facial expression (M (SD) proportion of trial) | Control | 0.01 (0.03) | 0.00 (0.00) | 0.01 (0.03) | 0.01 (0.04) | 0.00 (0.00) | 0.00 (0.00) |
| | Experimental | 0.04 (0.07) | 0.13 (0.12) | 0.11 (0.17) | 0.06 (0.11) | 0.06 (0.08) | 0.15 (0.20) |
| N (%) of infants showing concerned facial affect | Control | 1 (2.78) | 0 (0.00) | 2 (5.71) | 2 (4.55) | 0 (0.00) | 0 (0.00) |
| | Experimental | 14 (36.84) | 19 (54.29) | 15 (44.12) | 11 (25.00) | 15 (42.86) | 20 (47.62) |
| N (%) of infants showing comforting | Control | 0 (0.00) | 1 (2.86) | 0 (0.00) | 1 (2.27) | 2 (5.26) | 0 (0.00) |
| | Experimental | 6 (15.79) | 13 (37.14) | 4 (11.76) | 4 (8.70) | 10 (25.00) | 3 (7.14) |

tables (S7a–g in S1 File). The supplementary materials also include Bayesian equivalents of all generalised linear mixed models (S9 in S1 File), the results of which mirror the pattern of the frequentist models presented below.

### Condition effect

**Concerned facial expression.** The full-null model comparison was significant ($\chi^2(5) = 30.94$, $p < .001$) but the comparison to a reduced model with no interactions was not ($\chi^2(2) = 3.05$, $p = .217$). Single term deletions using drop1 showed that across timepoints and sites, infants showed more concerned facial expressions (proportional to trial length) in the experimental than in the control condition (estimate±SE = 0.49±0.09, $\chi^2(1) = 26.51$, $p < .001$, odds ratio (OR) = 1.62). Tables for all models with full model estimates, standard errors, and confidence intervals for all models and a post-hoc power analysis of significant effects can be found in the supplementary materials (S7a–g and S8 in S1 File).

**Comforting.** The full-null model comparison was significant ($\chi^2(5) = 42.56$, $p < .001$) but the reduced model comparison (with predictors entered as independent fixed effects and not as an interaction) was not ($\chi^2(2) = 1.92$, $p = .383$). Across timepoints and sites, infants were more likely to comfort in the experimental than in the control condition (estimate±SE = 2.47±0.53, $\chi^2(1) = 36.88$, $p < .001$, OR = 11.88). For full model estimates, standard errors, and confidence intervals, see S7b Table in S1 File.

### Age and site effects in the experimental condition

**Concerned facial expression.** The full-null model comparison did not reach significance ($\chi^2(3) = 3.89$, $p = .274$), indicating that Ugandan and UK infants spent similar proportions of the trials displaying concerned facial expression, with no change from 9 to 18 months. For full model estimates, standard errors, and confidence intervals, see in S7c Table in S1 File.

**Comforting.** The full-null model comparison was significant ($\chi^2(3) = 10.83$, $p = .013$) but the reduced model comparison was not ($\chi^2(1) = 0.01$, $p = .909$). Infants' likelihood to comfort increased significantly with age (estimate±SE = 1.21±0.43, $\chi^2(1) = 8.70$, $p = .003$, OR = 3.34; see Fig 1) but there was no significant effect of site (estimate±SE = -0.59±0.41, $\chi^2(1) = 2.12$, $p = .145$, OR = 0.55). For full model estimates, standard errors, and confidence intervals, see S7d Table in S1 File.

### Familiarity and site effects in the experimental condition

**Concerned facial expression.** The full-null model comparison did not reach significance ($\chi^2(3) = 1.43$, $p = .699$), indicating that infants at both sites showed similar amounts of concerned facial expression towards their mothers and the experimenters. For full model estimates, standard errors, and confidence intervals, see S7e Table in S1 File.

**Comforting.** The full-null model comparison was significant ($\chi^2(3) = 13.11$, $p = .004$) but the reduced model comparison was not ($\chi^2(1) = 0.00$, $p = .981$). Infants' likelihood to comfort was significantly higher for mothers than for experimenters (estimate±SE = 1.50±0.48, $\chi^2(1) = 11.44$, $p < .001$, OR = 4.47; see Fig 2) but there was no effect of site (estimate±SE = -0.55±0.43, $\chi^2(1) = 1.63$, $p = .202$). For full model estimates, standard errors, and confidence intervals, see S7f Table in S1 File.

### Facial concern predicting later comforting

The full-null model comparison did not reach significance ($\chi^2(3) = 3.74$, $p = .291$), indicating that concerned facial affect at 9 months did not predict comforting at 18 months. For full model estimates, standard errors, and confidence intervals, see S7g Table in S1 File.

### Effect of verbal cues

The distribution of infants who comforted before and after the explicit verbal cue of injury was given varied significantly with site ($\chi^2(1) = 5.99$, $p = .014$, OR = 7.33). Of the children who comforted either their mother or an experimenter at 18

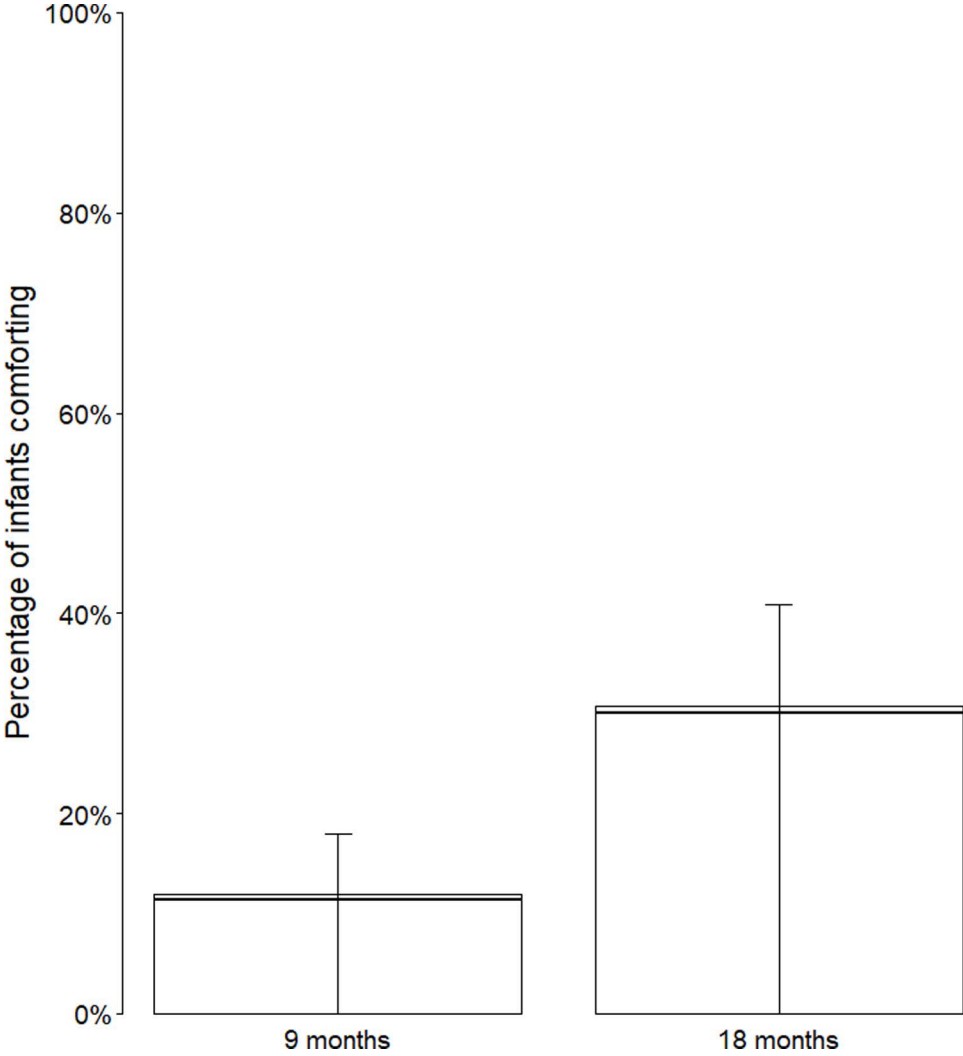

**Fig 1. Comforting of mother by infant age across sites.** Comforting of mother by age (9, 18 months) across sites (Uganda, UK) (Fig 1). Collapsed across sites given there were no detected site differences. Horizontal lines represent model estimates and whiskers represent 95% confidence intervals around model estimates.

months, most children from the UK site (77%) did so *before* the model explicitly labelled their distress, whereas the majority of children from the Ugandan site (69%) comforted *after* this verbal cue (see Fig 3).

## Discussion

Until recently, the early development of empathy had been understudied and as a result, theoretical frameworks may have underestimated its onset during the first year of life [37,38]. However, previous studies [7–9] have indicated that infants during the first year of life show two markers of empathy: expressions of concern and hypothesis testing, with some evidence also of comforting. Here, we expand on these findings with evidence across two culturally different settings to provide evidence of not only empathic concern but also comforting at 9 months. Although still rare in our cross-cultural sample, the detection of comforting behaviour at 9 months at both sites which is considered a reliable marker of empathy, is particularly notable, and supports the occurrence of comforting in the first year of life.

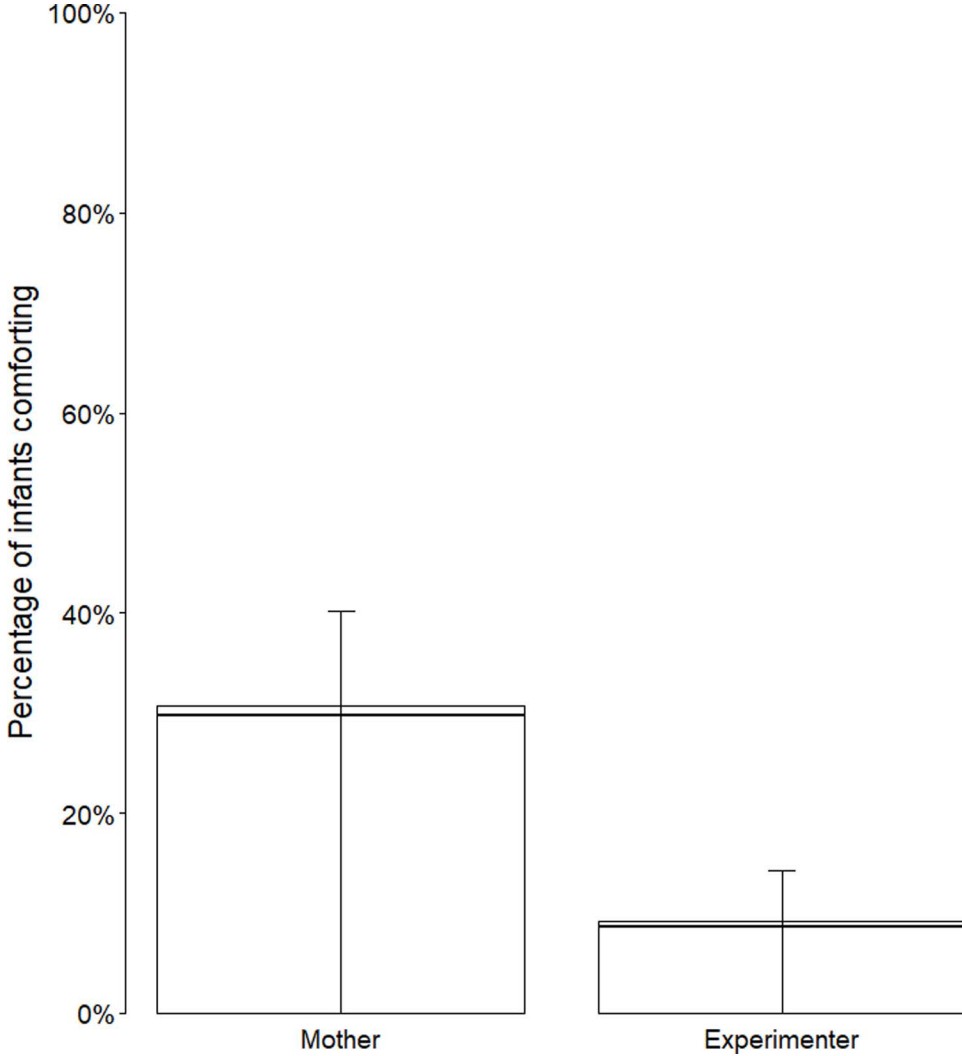

**Fig 2. Infant comforting at 18 months by model familiarity across sites.** Comforting by model familiarity (mother, experimenter) across sites (Uganda, UK) at 18 months (Fig 2). Collapsed across sites given there were no detected site differences. Horizontal lines represent model estimates and whiskers represent 95% confidence intervals around model estimates.

A key goal of our study was to provide a richer understanding of early empathy development that goes beyond a Western-centric model that has persisted in developmental science, but also in psychology more generally [39]. Despite differences in socio-cultural contexts and maternal socialisation attitudes between the two sites we sampled [27], we found similarities in infants' empathic behaviours. This apparent continuity complements previous work [17] which also found no differences in empathic behaviour in 19-month-old infants from Germany and India. While such findings cannot demonstrate universal patterns in empathy development, they do show that these behaviours show similarities in sites that differ in child-rearing practises and caregiver attitudes.

In line with previous studies [7,9], the number of infants who engaged in comforting increased from 9 to 18 months. In our sample, approximately a third of infants at 18 months showed at least one comforting behaviour in response to a distressed model, with most infants expressing this through patting or hugging the distressed adult. This supports the notion that comforting becomes more common and robust in the second year of life.

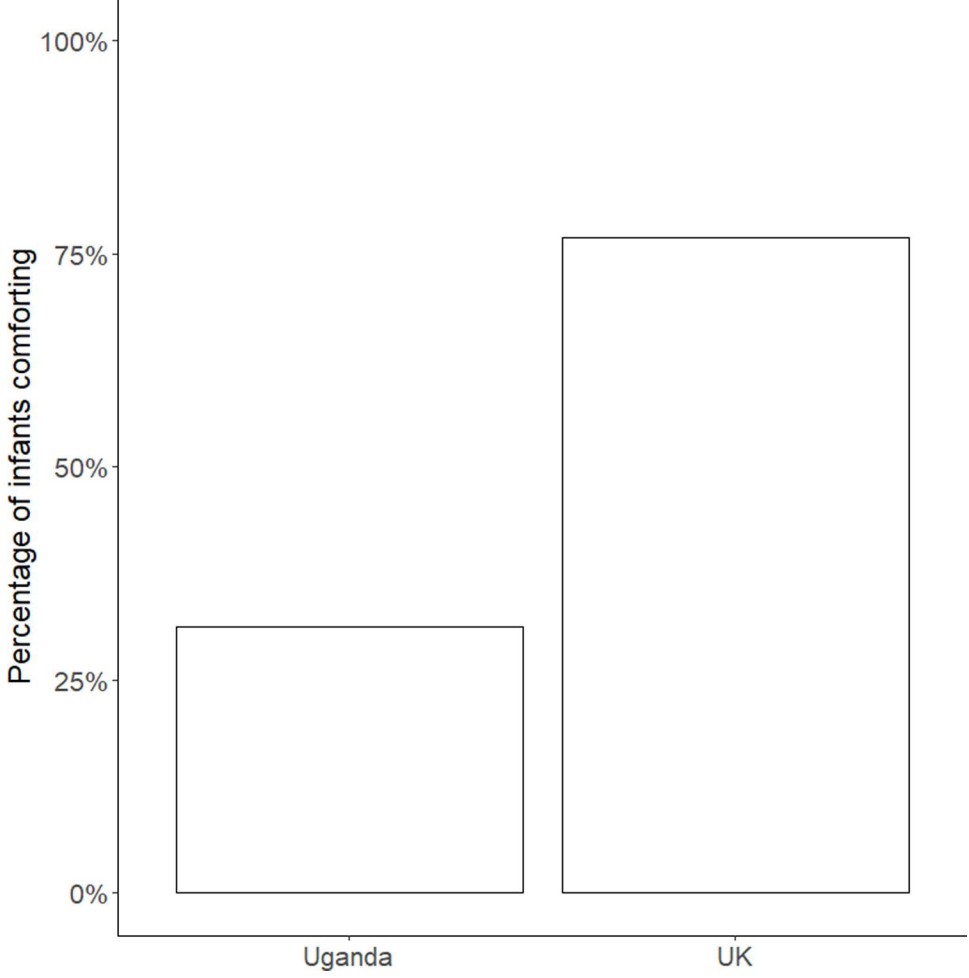

**Fig 3. Percentage of infants who comforted before the explicit verbal cue at 18 months by population.** Percentage of infants who comforted before the verbal cure, out of those infants who comforted at 18 months (*N* = 30), not the total sample.

In contrast to comforting, the production of facial expressions of concern was stable across both age points at both sites in our study. Previous studies have revealed a mixed picture of variation in facial expressions of concern in early development: Some [7] found no increase between the first and second year of life, whereas others [9] showed modest growth slopes during the first year of life which became weaker over the course of the second year. Yet others, [29] found facial concern increased from 14 to 36 months in a US sample, with the biggest change occurring from 14 to 20 months, indicating an increase over time which slows down with age. While concerned facial expressions in these previous studies were measured with a holistic rating of duration and intensity of facial concern, we decided to base our ratings on relative duration of concerned facial expression only. We decided against combining duration with intensity because facial expressions of emotions can vary with cultural norms around expressiveness [32], which may also affect the intensity with which they are expressed. Thus, whilst the relative duration of concerned facial expressions was stable from 9 to 18 months in our study, the intensity may have varied across timepoints or sites.

Since our procedure was based on prior work [11] which deviates slightly from the comforting paradigm used by other studies (e.g., [7,9,33]), our study took a slightly different approach to the adult's distress simulation (including gaze exchanges with the infant). We focussed on the occurrence of at least one comforting event and as explained above,

opted for duration without the inclusion of intensity when coding concerned facial affect. Despite these methodological differences, our results are in line with previous research (e.g., [7,9,29]) conducted in the US and Israel. This supports the apparent robustness of the presence of concerned facial expressions and comforting in early development, despite methodological adaptations and cross-cultural variability in caregiving practices.

Although our main analysis focused on the experimental condition, we also statistically compared the control and experimental conditions. We found that infants were significantly more likely to comfort the model in the experimental compared to the control condition, and that in the experimental condition they also displayed concerned facial affect for a greater proportion of the trial. The descriptive data (Table 1) show that very few infants engaged in these behaviours in the control condition. These findings indicate that it is highly unlikely that comforting behaviours and concerned facial affect in the experimental condition were a response to encountering an unresponsive/ distracted adult. Although it has been shown that 18-month-olds show concern and comforting to an adult who has injured themselves but does not act as if in pain [28], we found little evidence that infants in our sample behaved in this way. This suggests that while infants may be able to infer emotional needs after seeing causes for distress [28], they are also sensitive to communicative cues about emotional needs, such as vocalisations of pain and visual attention and rubbing gestures on the injured body part and respond empathically to such cues.

By using a longitudinal approach, we could assess whether early empathy predicted empathic responding during the second year of life. In our study, concerned facial expressions at 9 months did not predict comforting at 18 months. These results contribute to a mixed picture of how empathy develops: On one hand, prior work has found that concerned facial expressions at 8 months predicted empathic comforting at 12 months but not at 14 months [7]. On the other hand, others have found that concerned facial expressions at 6 months (but not at 3 or 12 months) predicted comforting at 18 months [9], or that different developmental trajectories of early facial expressions of concern predicted social competence at 36 months [10]. While these studies used intensity ratings of concern and comforting, we used a duration-based approach for facial expressions of concern. Although this accounted for potential variation in cultural norms for expressing emotions such as concern, it may have contributed to the lack of a predictive relationship between the two. More research is needed to understand how different behavioural markers of empathetic behaviour relate to each other and how stable individual variation in early empathetic behaviours is over development, accounting for socio-cultural influences on the expression of both concern and comforting.

In line with previous studies (e.g., [29]), our findings on comforting at 18 months support the social bias of empathy, whereby empathy is evolutionarily biased towards kin and other individuals perceived as similar or familiar [2]. In our study, infants were more likely to comfort their mother than an experimenter, regardless of population, however facial expressions of concern were comparable regardless of the identity of the adult model, mirroring previous studies. It is possible that reduced comforting towards an unfamiliar adult may also be explained by other factors, such as being more reluctant to approach or interact with them. Future research examining underlying psycho-physiological mechanisms is needed to address this question further.

Although we found many similarities in the overall occurrence of empathy behaviours, our study revealed some interesting site level variation. Compared to Ugandan infants, the UK infants in our sample were more likely to comfort before a verbal cue to the injury was provided ("Look, I hurt myself!"). Although this result is based on a modest number of infants who comforted at 18 months ($N = 30$), it suggests variation in the occurrence of spontaneous comforting. Generally, empathy is conceptualised as being spontaneous [40], occurring because of the observer sharing a target's emotional state, without the need for the emotional state to be verbally labelled. However, it may be that verbal cues to distress are used differently across socio-cultural settings and expectations of how others respond to those cues are culturally shaped. Early anthropological research with Ugandan agriculturalists described a tendency toward more regulated expressions of emotions and fewer affective exchanges between caregivers and infants [26,27]. This observation overlaps with findings from Holden et al. [24] that Ugandan mothers were more likely to prioritise relational attitudes (emphasising compliance

and obedience) than UK mothers from these sites. These socialisation practices may contribute to the Ugandan infants responding more readily to explicit labels of emotions rather than engaging in spontaneous comforting. However, as we did not directly assess specific caregiving practices and their link with infant comforting behaviours, this possibility remains speculative. Future research should address this limitation by investigating local cultural norms around affective expression and socialisation practices, such as frequency of emotional exchanges or scaffolding of empathic behaviours, at each study site. Assessing these aspects alongside infants' comforting behaviours could clarify how different cultural approaches to expressing and responding to emotions shape infants' empathic development.

An additional future direction would be the inclusion of measures of parental empathy at the individual level, going beyond cultural generalisations: In samples from North America, Australia, China, and the Netherlands, parents' empathic tendencies and abilities have been found to positively correlate with those of their offspring at various stages of development (e.g., [41–44]) and that parental empathy is itself affected by the socio-cultural environment of the parent. In this study, we did not assess individual mothers' empathy, in part due to the challenges of assessing parental attitudes using self-report in a low-literacy environment. However, future research examining the influence of both group-level and individual-level factors on empathy development would aid a more in-depth understanding of socialisation influences on empathy.

Taken together, our findings provide support for the presence of empathic concern and comforting in infancy. However, using behavioural markers of such a complex process as empathy is not without challenges. Concerned facial expressions can be subtle and may be similar to other forms of negative affect. Although we addressed this by excluding co-occurrence with signs of distress, such as crying or whimpering, an important future direction would be to validate concerned facial expressions coding using systematic methods, like the Facial Action Coding System (FACs) [45]. Similarly, comforting behaviour may have alternative explanations, such as seeking reassuring contact in response to someone else's distress. Although visible signs of infant distress, which may have led participants to seek out physical touch with their mother, were as rare in our study as in others (e.g., [7,9]), it would be beneficial for future research to supplement behavioural measures with physiological measures, like eye-tracking. Eye tracking has been successfully used in the realm of instrumental helping to capture understanding of others' needs or intrinsic motivation to help (e.g., [46,47]), and a similar approach could be taken to comforting.

The extent of alloparental care represents a striking difference between our samples from Uganda and the UK: Most infants in our UK sample were primarily cared for by their mother (especially during the first year while on maternity leave) and their father. In our Ugandan sample, caregiving was distributed between a larger number of people, including other members of the household, neighbours, and siblings [24]. In this study, we focused on mothers to aid comparability with previous research [e.g., 7–9] however, future work should also consider the role of other caregivers, including fathers and where ecologically relevant, non-parental caregivers.

A broader constraint of our study was the use of an empathy paradigm originally designed for Western urbanised/industrialised settings, and the experimental situation was likely more novel to the Ugandan participants. We attempted to mitigate this novelty and ensure that the paradigm was culturally appropriate by piloting, co-producing, and conducting it with local Ugandan researchers. By the first timepoint of this study, all mother-infant dyads at both sites had already participated in two previous timepoints as part of the broader longitudinal project, thus reducing the novelty of the situation for all participants. If, despite our efforts, differences in the ecological validity of the task remained across our two study sites, then it is possible that the current study underestimated the likelihood of Ugandan infants to comfort or show concerned facial affect, but we can be confident that early empathic behaviours in our Ugandan sample were at least as common as in our UK sample.

In conclusion, by taking a rigorous longitudinal approach in two different cultural contexts, our study supports the view that multiple behavioural markers of empathy start to emerge in the first year of life. Infants from both sites showed similar patterns of empathy-related behaviour, being more likely to comfort their mother compared to an unfamiliar experimenter and more likely to engage in comforting with age. The only key socio-cultural difference that emerged was that the UK infants were more likely than Ugandan infants to offer spontaneous comfort before the model gave an explicit verbal

cue. Overall, our results highlight the early emergence of behavioural markers of empathy at two sites with differences in socio-cultural settings.

## Supporting information

**S1 File. S1–S10 supplementary materials** . Including coding scheme, full model outcomes, and additional analyses. (DOCX)

## Acknowledgments

We are very grateful to the families and communities in Uganda and the UK that have participated in this research. We thank the directors and team at Budongo Conservation Field Station for supporting our research in Uganda, as well as Prof Robert Newton for his support and advice. We thank Dr Kirsty Graham for their significant contribution to the conceptualisation, methodology, investigation and writing of this study. In Uganda, we are grateful to research assistants Emma Sopelsa-Hall, John McCutcheon, Charlie Ives and the late Hellen Biroch who assisted with data collection. Thank you to Businge Ambrose, Anguzu Julius, Masereka Brenda, Monday Mbotella and Mua Simon for translation of materials. In the UK, we are grateful to research assistants Rebecca Anderson, Molly Bowns, Lucy Dunn, Megan Earl, Keshy Emmanuel, Harold Green, Charlie Ives, Harshanaa Patel, Rhiannon Pearce, Maddy Peryer, Barbora Sodomkova, Emma Standley, Sonnie Tan, Maisie Thurman, Joseph Vogliqi, Daisy Whitwood, and Caitlin Woods, who assisted with data collection.

## Author contributions

**Conceptualization:** Carlo Vreden, Joanna C. Buryn-Weitzel, Eve Holden, Nicole J. Lahiff, Katie E. Slocombe, Zanna Clay.

**Formal analysis:** Carlo Vreden, Bahar Tuncgenc.

**Funding acquisition:** Katie E. Slocombe, Zanna Clay.

**Investigation:** Carlo Vreden, Joanna C. Buryn-Weitzel, Santa Atim, Ed Donnellan, Maggie Hoffman, Eve Holden, Michael Jurua, Charlotte V. Knapper, Sophie Marshall, Josephine Paricia, Florence Tusiime, Claudia Wilke, Katie E. Slocombe.

**Methodology:** Carlo Vreden, Joanna C. Buryn-Weitzel, Santa Atim, Ed Donnellan, Eve Holden, Michael Jurua, Nicole J. Lahiff, Josephine Paricia, Bahar Tuncgenc, Florence Tusiime, Claudia Wilke, Katie E. Slocombe, Zanna Clay.

**Project administration:** Joanna C. Buryn-Weitzel, Ed Donnellan, Maggie Hoffman, Eve Holden, Charlotte V. Knapper, Nicole J. Lahiff, Sophie Marshall, Claudia Wilke, Katie E. Slocombe.

**Supervision:** Katie E. Slocombe, Zanna Clay.

**Validation:** Carlo Vreden.

**Visualization:** Carlo Vreden.

**Writing – original draft:** Carlo Vreden, Joanna C. Buryn-Weitzel, Katie E. Slocombe, Zanna Clay.

**Writing – review & editing:** Carlo Vreden, Joanna C. Buryn-Weitzel, Santa Atim, Ed Donnellan, Maggie Hoffman, Eve Holden, Michael Jurua, Charlotte V. Knapper, Nicole J. Lahiff, Sophie Marshall, Josephine Paricia, Bahar Tuncgenc, Florence Tusiime, Claudia Wilke, Katie E. Slocombe, Zanna Clay.

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
