## [Decision Letter · Decision Letter 0]

5 Nov 2024

PONE-D-24-26181Early empathy development: concern and comforting in 9- and 18-month-old infants from Uganda and the UKPLOS ONE

Dear Dr. Vreden,

Thank you for submitting your manuscript to PLOS ONE. After careful consideration, we feel that it has merit but does not fully meet PLOS ONE’s publication criteria as it currently stands. Therefore, we invite you to submit a revised version of the manuscript that addresses the points raised during the review process.

**ACADEMIC EDITOR: **Thank you for submitting your manuscript to PLOS ONE and I appreciate your patience while awaiting the decision. I am delighted to have received reviews from two experts in the field, and I have also reviewed the manuscript with great interest. As you will see from the comments, the reviewers see considerable merit in your paper and I agree with their assessment. The reviewers also have some concerns. I will not repeat their comments here, but I would like to ask you to consider and address each comment carefully. In addition, I was wondering if it would be beneficial to include Bayesian analyses (or equivalence testing), particularly for the null effects you report, to determine whether there is evidence supporting these null findings or the evidence is inconclusive. I understand this may require additional work, but as your study offers a unique opportunity to examine the ontogeny of empathy across cultures, I believe this could be a valuable addition.  Milica Nikolic==============================

We look forward to receiving your revised manuscript.

Kind regards,

Milica Nikolic, Ph.D

Academic Editor

PLOS ONE

Journal Requirements:

"European Research Council Consolidator Grant awarded to KS (724608) and European Research Council Starting Grant awarded to ZC (802979)"

"We are very grateful to the families and communities in Uganda and the UK that have participated in this research. We thank the directors and team at Budongo Conservation Field Station for supporting our research in Uganda. This research received ethical approval from the University of York Psychology Ethics Committee, the Ethics Sub-Committee of the Department of Psychology, Durham University, the Uganda Virus Research Institute (UVRI-045/2017) and the Ugandan National Council for Science and Technology (Ref://SS4545). This research was funded by the European Research Council Consolidator Grant awarded to KS (724608) and the European Research Council Starting Grant awarded to ZC (802979)."

"European Research Council Consolidator Grant awarded to KS (724608) and European Research Council Starting Grant awarded to ZC (802979)"

5. We note that Figures 1 (a and b) in your submission contain copyrighted images. All PLOS content is published under the Creative Commons Attribution License (CC BY 4.0), which means that the manuscript, images, and Supporting Information files will be freely available online, and any third party is permitted to access, download, copy, distribute, and use these materials in any way, even commercially, with proper attribution. For more information, see our copyright guidelines: http://journals.plos.org/plosone/s/licenses-and-copyright.

a. You may seek permission from the original copyright holder of Figures 1 (a and b) to publish the content specifically under the CC BY 4.0 license. 

6. We note that Figures 1 (a and b) includes an image of a patient/participant in the study. 

Reviewers' comments:

Reviewer's Responses to Questions

**Comments to the Author**

1. Is the manuscript technically sound, and do the data support the conclusions?

Reviewer #1: Yes

Reviewer #2: Yes

2. Has the statistical analysis been performed appropriately and rigorously? 

Reviewer #1: Yes

Reviewer #2: Yes

3. Have the authors made all data underlying the findings in their manuscript fully available?

Reviewer #1: Yes

Reviewer #2: Yes

4. Is the manuscript presented in an intelligible fashion and written in standard English?

Reviewer #1: Yes

Reviewer #2: Yes

5. Review Comments to the Author

Reviewer #1: I believe this paper is of high quality and will be of great interest to the broad readership of this journal. My comments primarily focus on clarifying certain data analysis steps and specific aspects of the experimental procedure. If these comments are addressed, I would be inclined to accept this work.

The work by Vreden and colleagues aimed to investigate the universality of early signs of empathy across two testing sites: the UK and Uganda. This research has been rigorously conducted and is much needed, both to enhance our understanding of social development and to address the issue that most of our knowledge on infant development comes from Western studies. I have a few comments that I hope will improve the quality of the paper and facilitate easy understanding for readers.

- Throughout the entire introduction, particularly on pages 6 and 7, the work would benefit from a clearer emphasis on the research gap. The authors describe several studies in which social development has been tested in different locations, but what does this study add to what is already known? On page 6, the authors mention, 'Given the need for infancy research in more diverse cultures with varied parenting practices,' but a deeper justification is needed beyond simply testing empathy in different cultures. Additionally, while reading the introduction, it seems that the authors are planning to examine the effect of different sociocultural environments on early empathy (for example, by exploring correlations between parenting practices and signs of empathy). However, in reality, early signs of empathy are tested at two sites without directly investigating whether variables specific to these environments impact empathy.

- In addition of the table SI1, I would add numbers for participants per site – how, any participants have been tested? How many of these completed the task? How many of these came back for visit 2? How many of these were excluded and if so why? I think this would be a very useful resource for researchers who would like to carry out similar research project.

- Was any power analysis performed to calculate the total sample needed? If not, how N=93 (divided between the two samples) was decided?

- It is unclear why the scenario with the mum being injured changed between 9 and 18 mo and with the experimenter not. But also above the authors said that that the 9-month-olds were not tested with the experimenter condition. I think this whole section should be made clearer, maybe even with a graphical representation?

- The choice of the control condition is somewhat unclear, as the model still injures herself in this scenario but does not react to it. Would this not appear strange to the child? Would it not have been better to include a scenario where the model seeks help (e.g., drops a pen out of reach) without being injured? This would better account for the child’s general tendency to help. In either case, it would be beneficial to provide more justification for why this was considered an appropriate control condition, as it would be valuable for other researchers interested in using a similar task.

- How the authors made sure the mothers’ ‘acting skills’ were similar across all mothers? The experimenter presumably does the same scene for every testing session but the mother not. Have there not been any instances in which mothers failed to act, for example started laughing? This are other useful information for setting up a similar study.

- Through the whole set of statistical analyses, why the authors considered the trails in total as dependent variable? I would have expected that a ‘comforting’ value towards the mother and one toward the experimenter would have been calculated per each infant averaged across trials , so to use this one as dependent variables in statistical analyses?

- Error bars in fig 4 are missing.

Reviewer #2: Dear Authors,

Thank you for the opportunity to review this manuscript, which examines the early development of empathy-related behaviours in 9- and 18-month-old infants from Uganda and the UK. The manuscript is well-written and reads very clearly. The study offers valuable insights into empathy development by analyzing concern and comfort behaviours separately, in response to an adult's pain simulation— a commonly employed task for measuring empathy development in young children. The cross-cultural, longitudinal design, inclusive of both experimental and control conditions, as well as repeated measures across adult identities, represents important methodological strengths often underexplored in similar research. Additionally, the use of naturalistic observations within the family home, which supports the ecological validity of the collected data, is a significant contribution to the field.

I have a few, minor questions and suggestions for improving the manuscript:

Method section:

Information regarding the socioeconomic status (SES) and professions of the sample could be of interest. Even though the focus is on the cross-cultural aspect, and in Uganda, mothers’ professions may be limited with less variation than in the UK, such information could still be relevant, as longitudinal studies in infancy are rare, and this data could be a beneficial foundation for future studies in empathy research.

On page 12, the text states: "Each trial began with the infant sitting on the floor, either on their own or, if necessary, supported by an adult who was not part of the task." Could you clarify how often this support was required?

Page 14. The authors appropriately used Cohen’s kappa for frequency behaviours and ICC for the duration of behaviours, as these metrics are well-suited to the respective data types. It might be helpful to briefly clarify the methodological decision of employing different interrater reliability coefficients (for categorical and continuous data, respectively).

The authors present data from 44 infants from Uganda and 49 from the UK, resulting in a total of 465 valid control and experimental trials (which I estimate to involve approximately 230 observations for the experimental condition). Given this sample size, it would be beneficial to include a power analysis to confirm that the study is adequately powered for the analyses conducted. Doing so would enhance the robustness and transparency of the findings.

Results section:

While I understand there may not be correlations between infant empathic behaviours with mothers and experimenters (p. 17: "there was no case an infant at 18 months comforted both their mother and experimenter"), I wonder if there were correlations between empathy-related behaviours across age. For example, was concern and/or comfort at 9 months associated with the same measure at 18 months? Providing information about the temporal stability of early-developing empathy-related behaviours would be highly valuable, as the field currently has a scarcity of longitudinal studies on empathy in infancy. I would therefore encourage the authors to include a correlation table for all the experimental variables.

Supplementary S7 presents many tables. Please distinguish the tables clearly (e.g., Table S7a, Table S7b) and ensure they are referenced in the main text when presenting the results. For example, in the section on "age and site effects," alongside the mention of Figure 2, you could also refer to the specific table in S7 where the corresponding estimates are presented, to provide clearer guidance for readers. The same suggestion applies when presenting results that reference Figure 3. (Currently, supplementary Table S7 is mentioned in the manuscript in relation to “condition effects”, but no such tables appear in the supplemental material).

The note in all tables of S7 (i.e., "Not indicated because of having a very limited interpretation") is unclear. Could you provide more context for this? If the result holds little value or could mislead readers, it may be more appropriate to omit it. Providing a clearer explanation would improve transparency.

One of the tables in S7 misses the column for the df.

I suggest restructuring Table S7 to align with the results presentation described in the manuscript on page 16 and in Table 1. This adjustment would help readers follow the results more easily in the supplementary materials.

Figures 2, 3, and 4 could benefit from additional details to clearly convey the results at a glance, even for readers who haven't thoroughly gone through the manuscript. For instance, in Figure 4, it would be helpful to indicate that the percentages represent comforting behaviours prior to any verbal cues. For Figure 3, could be useful to specify that the infant comforting behaviours with mother and experimenter were observed at 18 months.

Discussion section:

While the authors have made significant strides in examining familiarity effects by observing infant empathic responses to mothers and experimenters, the paternal figure has not been included in the study design. I recommend acknowledging this as a limitation and suggesting it as an area for future research.

The importance of parental empathy for socialization practices could be discussed. Prior research consistently indicates associations between parental empathy and the development of empathy in their offspring (e.g., Eisenberg et al., 1991; 2020; Farrant et al., 2012; Salvadori et al., 2021; Soenens et al., 2007; Volling et al., 2008). Did the authors collect any measures of parental empathy? Addressing this aspect in the discussion could highlight it as both a limitation and a direction for future research.

On page 26, please adjust the text “to validation” to “to validate” or “the validation.”

References:

Eisenberg, N. (2020). Findings, issues, and new directions for research on emotion

socialization. Developmental Psychology, 56(3), 664–670.

Eisenberg, N., Fabes, R. A., Schaller, M., Carlo, G., & Miller, P. A. (1991). The relations of parental

characteristics and practices to children’s vicarious emotional responding. Child

Development, 62(6), 1393-1408.

Farrant, B. M., Devine, T. A., Maybery, M. T., & Fletcher, J. (2012). Empathy, perspective taking and

prosocial behaviour: The importance of parenting practices. Infant and Child

Development, 21(2), 175-188.

Salvadori, E. A., Colonnesi, C., Vonk, H. S., Oort, F. J., & Aktar, E. (2021). Infant emotional mimicry

of strangers: associations with parent emotional mimicry, parent-infant mutual attention,

and parent dispositional affective empathy. International Journal of Environmental Research

and Public Health, 18(2), 654.

Soenens, B., Duriez, B., Vansteenkiste, M., & Goossens, L. (2007). The intergenerational

transmission of empathy-related responding in adolescence: The role of maternal

support. Personality and Social Psychology Bulletin, 33(3), 299-311.

Volling, B. L., Kolak, A. M., & Kennedy, D. E. (2008). Empathy and compassionate love in early

childhood: Development and family influence. In B. Fehr, S. Sprecher, & L. G. Underwood

(Eds.), The science of compassionate love: Theory, research, and applications (pp. 161-200).

Wiley-Blackwell.

6. PLOS authors have the option to publish the peer review history of their article (what does this mean? ). If published, this will include your full peer review and any attached files.

**Do you want your identity to be public for this peer review?** For information about this choice, including consent withdrawal, please see our Privacy Policy .

Reviewer #1: No

Reviewer #2: No

---

## [Author Response · Author response to Decision Letter 1]

12 Dec 2024

(This response has also been uploaded as a file, titled Response to Reviewers.)

Response to Reviewers

Academic Editor:

Thank you for submitting your manuscript to PLOS ONE and I appreciate your patience while awaiting the decision. I am delighted to have received reviews from two experts in the field, and I have also reviewed the manuscript with great interest.

As you will see from the comments, the reviewers see considerable merit in your paper and I agree with their assessment. The reviewers also have some concerns. I will not repeat their comments here, but I would like to ask you to consider and address each comment carefully. In addition, I was wondering if it would be beneficial to include Bayesian analyses (or equivalence testing), particularly for the null effects you report, to determine whether there is evidence supporting these null findings or the evidence is inconclusive. I understand this may require additional work, but as your study offers a unique opportunity to examine the ontogeny of empathy across cultures, I believe this could be a valuable addition.

Thank you very much for your consideration of our manuscript and your positive feedback on our study. We appreciate your suggestion of additional Bayesian analyses in particular to examine the null results, we have consequently added Bayesian equivalents of the main models in our analysis to the supplementary materials. These additional analyses follow the pattern of our existing results, showing credible intervals not crossing zero and high probability of direction for those effects which were also significant in the frequentist models. Regarding the effects which were not significant in our frequentist analysis, the Bayesian models show credible intervals crossing zero for all of these and probabilities of direction below the 97.5% criterion considered to be equivalent to a frequentist p-value of 0.05. Although we cannot test for support in favour of the null hypothesis for these effects (because Bayes Factors which would normally be used for this test cannot be applied to GLMMs), the convergence of results across both the frequentist and Bayesian analysis suggests good robustness of our findings (see supplementary materials S9). We have now highlighted these Bayesian models in the statistical analysis section (lines 417-419) and summarised that in the results the pattern obtained from the Bayesian models mirrored those of the frequentist ones, giving confidence in our pattern of results (lines 435-437).

Reviewer #1:

I believe this paper is of high quality and will be of great interest to the broad readership of this journal. My comments primarily focus on clarifying certain data analysis steps and specific aspects of the experimental procedure. If these comments are addressed, I would be inclined to accept this work.

The work by Vreden and colleagues aimed to investigate the universality of early signs of empathy across two testing sites: the UK and Uganda. This research has been rigorously conducted and is much needed, both to enhance our understanding of social development and to address the issue that most of our knowledge on infant development comes from Western studies. I have a few comments that I hope will improve the quality of the paper and facilitate easy understanding for readers.

We are grateful for this encouragement and positive global assessment for our work – thank you!

- Throughout the entire introduction, particularly on pages 6 and 7, the work would benefit from a clearer emphasis on the research gap. The authors describe several studies in which social development has been tested in different locations, but what does this study add to what is already known? On page 6, the authors mention, 'Given the need for infancy research in more diverse cultures with varied parenting practices,' but a deeper justification is needed beyond simply testing empathy in different cultures. Additionally, while reading the introduction, it seems that the authors are planning to examine the effect of different sociocultural environments on early empathy (for example, by exploring correlations between parenting practices and signs of empathy). However, in reality, early signs of empathy are tested at two sites without directly investigating whether variables specific to these environments impact empathy.

Thank you for highlighting this. We have now expanded our conclusions in the introduction paragraph detailing previous cross-cultural work to more clearly highlight the current gaps in our knowledge (lines 111-122):

We then explicitly acknowledge that although the variation between our sites in empathy-relevant factors provides a rationale for comparing these two cultural contexts, examining their influence was beyond the scope of the current study. We hope this will set reader expectations to be more accurate (lines 140-143). To confirm, the aim of our study was to lay the groundwork of establishing cross-cultural similarities and differences in early empathy development, in order to then, in a second step, conduct a more in-depth analysis of how possible variation might be linked with specific cultural practices. We acknowledge that this is a limitation and have added this in the discussion section (lines 600-610).

- In addition of the table SI1, I would add numbers for participants per site – how, any participants have been tested? How many of these completed the task? How many of these came back for visit 2? How many of these were excluded and if so why? I think this would be a very useful resource for researchers who would like to carry out similar research project.

Thank you for highlighting this. We have now added how many participants were tested at each timepoint and site, before application of exclusion criteria, to table S1. Additionally, the text accompanying table S1 in the supplementary material now contains information on how many trials were excluded for what reason and participant retention from visit 1 to 2.

- Was any power analysis performed to calculate the total sample needed? If not, how N=93 (divided between the two samples) was decided?

We now explain on lines 205-209 that this study’s sample size was constrained by the sample size and dropout rate of the larger longitudinal project which this study was part of. Therefore no specific a priori power analysis was conducted for this particular study. However, the supplementary materials (S8) now include a post-hoc power analysis for all detected significant effects, all of which show sufficient power. As per the editor’s suggestion, we now also report additional Bayesian analyses in the supplementary materials (S9), which follow the pattern of our frequentist results. This convergence across statistical approaches suggests robustness of our findings.

- It is unclear why the scenario with the mum being injured changed between 9 and 18 mo and with the experimenter not. But also above the authors said that that the 9-month-olds were not tested with the experimenter condition. I think this whole section should be made clearer, maybe even with a graphical representation?

Thank you for pointing out that this needs clarification. We have now expanded on this in the design section (lines 257-277) to explain that it was only feasible for infants at 9 months to participate in a single trial (fatigue, concentration limitations), and that was done with the mother. In terms of the type of scenario used, as only 2 scenarios were viable to use in Uganda, it was necessary to repeat one of the scenarios across a child’s three trials (1 at 9 months and 2 at 18 months). To aid comparability across sites, we therefore ensured that in both sites the scenario used by the mother at 9 months was also used by the experimenter at 18 months and the mother at 18 months used a novel scenario. This clarification has been provided as noted above.

- The choice of the control condition is somewhat unclear, as the model still injures herself in this scenario but does not react to it. Would this not appear strange to the child? Would it not have been better to include a scenario where the model seeks help (e.g., drops a pen out of reach) without being injured? This would better account for the child’s general tendency to help. In either case, it would be beneficial to provide more justification for why this was considered an appropriate control condition, as it would be valuable for other researchers interested in using a similar task.

Thanks for this point. Our control condition was based on that used by Dunfield et al. (2011) who also used an experimental condition which featured an injury followed by simulated distress and a control condition which still featured the injury but no distress. We know that 18-month-olds understand the causality of injuries and show empathic responses towards an adult after an injury even if the adult does not simulate distress (Vaish et al., 2009). Here, we were more interested in their understanding of and responding to distress signals specifically, as a genuine empathic response should be based on not only objective factors (like the occurrence of an injury) but also on correctly interpreting the injured individual’s expressed need for empathic comforting.

Another control condition used in previous literature does not feature any injury (e.g., Roth-Hanania et al., 2011; Davidov et al., 2021) but rather uses an unrelated neutral scenario. However, using such a completely different scenario makes it difficult to pinpoint what exactly may cause potential differences in infant behaviour between the two conditions.

We realise this could have been clearer and so we now elaborate on our justification for modelling our control condition on Dunfield’s and also acknowledge that different control conditions have been used in other previous studies in lines 161-168.

- How the authors made sure the mothers’ ‘acting skills’ were similar across all mothers? The experimenter presumably does the same scene for every testing session but the mother not. Have there not been any instances in which mothers failed to act, for example started laughing? This are other useful information for setting up a similar study.

Sufficient acting skills were indeed key for the success of the paradigm, and the reviewer is correct that the task was more challenging for the mothers who had more limited practice as compared to the experimenter. This is evidenced by some mother trials having to be excluded from further analysis for breaking character compared to none of the experimenter trials, these numbers can now be found broken down by culture in the supplementary material S1).

In our coding scheme, we had specific criteria for exclusion where mothers broke character. During the procedure itself, we tried to standardise the realism of the mothers’ pretend accidents and subsequent reactions by providing mothers with the opportunity to practice the procedure prior to the experiment and to receive feedback from the experimenter. Moreover, we provided mothers with auditory cues to remind them of what they should be doing throughout each trial. We have now inserted additional text on lines 290-295 to explain this in more detail.

- Through the whole set of statistical analyses, why the authors considered the trails in total as dependent variable? I would have expected that a ‘comforting’ value towards the mother and one toward the experimenter would have been calculated per each infant averaged across trials , so to use this one as dependent variables in statistical analyses?

We apologise that this was not clearer. We have now added details to lines 398-401 and to the supplementary materials S1 to explain that not all infants had valid trials with both adult partners at 18 months (one visit was missed due to illness, or both trials were completed, but one had to be excluded later). We now specify that all measures were extracted for each trial (e.g. separately for mother and experimenter trials). Whether infants were more likely to comfort their mother or an experimenter was also one of the questions we wanted to address with our analyses, so retaining separate scores for mother and experimenter trials rather than creating a composite measure was important.

- Error bars in fig 4 are missing.

Figure 4 (now Figure 3) presents the absolute number of infants who comforted before the explicit cue, i.e., raw data and not the mean by site. Therefore, this figure does not include error bars.

Reviewer #2:

Dear Authors,

Thank you for the opportunity to review this manuscript, which examines the early development of empathy-related behaviours in 9- and 18-month-old infants from Uganda and the UK. The manuscript is well-written and reads very clearly. The study offers valuable insights into empathy development by analyzing concern and comfort behaviours separately, in response to an adult's pain simulation— a commonly employed task for measuring empathy development in young children. The cross-cultural, longitudinal design, inclusive of both experimental and control conditions, as well as repeated measures across adult identities, represents important methodological strengths often underexplored in similar research. Additionally, the use of naturalistic observations within the family home, which supports the ecological validity of the collected data, is a significant contribution to the field.

We really appreciate this positive global assessment of our work – many thanks.

I have a few, minor questions and suggestions for improving the manuscript:

Method section:

Information regarding the socioeconomic status (SES) and professions of the sample could be of interest. Even though the focus is on the cross-cultural aspect, and in Uganda, mothers’ professions may be limited with less variation than in the UK, such information could still be relevant, as longitudinal studies in infancy are rare, and this data could be a beneficial foundation for future studies in empathy research.

Thanks for this suggestion. We agree this may be of interest, and so now report additional information on mothers’ employment and a mean parental Hollingshead SES value for the Ugandan and UK samples (lines 222-229).

On page 12, the text states: "Each trial began with the infant sitting on the floor, either on their own or, if necessary, supported by an adult who was not part of the task." Could you clarify how often this support was required?

Across sites, ca. 10% of infants required the presence of an additional adult. Note that this adult was only there to keep the infant in view of the cameras at the trial start, as some infants tried to crawl or walk away before their mother could start her distress simulation. As soon as the trial began, the infant would sit on their own and the adult was turned away and pretending to be busy with something else (e.g., looking at a questionnaire), in order to not be available as a potential interaction partner or comforter. We have also clarified this further in the manuscript (lines 298-304).

Page 14. The authors appropriately used Cohen’s kappa for frequency behaviours and ICC for the duration of behaviours, as these metrics are well-suited to the respective data types. It might be helpful to briefly clarify the methodological decision of employing different interrater reliability coefficients (for categorical and continuous data, respectively).

We have clarified the type of measure (categorical for comforting, continuous for concern) that reliability is being calculated for in this section (lines 346-347).

The authors present data from 44 infants from Uganda and 49 from the UK, resulting in a total of 465 valid control and experimental trials (which I estimate to involve approximately 230 observations for the experimental condition). Given this sample size, it would be beneficial to include a power analysis to confirm that the study is adequately powered for the analyses conducted. Doing so would enhance the robustness and transparency of the findings.

As explained above, an a priori power analysis was not possible due to this experiment being part of a larger longitudinal project with a fixed sample size. Therefore, we have now added a post-hoc power an

---

## [Decision Letter · Decision Letter 1]

18 Feb 2025

Early empathy development: concern and comforting in 9- and 18-month-old infants from Uganda and the UK

PONE-D-24-26181R1

Dear Dr. Vreden,

We’re pleased to inform you that your manuscript has been judged scientifically suitable for publication and will be formally accepted for publication once it meets all outstanding technical requirements.

Kind regards,

Milica Nikolic, Ph.D

Academic Editor

PLOS ONE

Additional Editor Comments (optional):

Dear Dr. Vreden,

I have now received reviews from the two original reviewers. Both reviewers were satisfied with how you addressed their comments. I agree with their assessment and am delighted to accept this paper for publication. Congratulations on this interesting work!

Dr. Milica Nikolic

Reviewers' comments:

Reviewer's Responses to Questions

**Comments to the Author**

1. If the authors have adequately addressed your comments raised in a previous round of review and you feel that this manuscript is now acceptable for publication, you may indicate that here to bypass the “Comments to the Author” section, enter your conflict of interest statement in the “Confidential to Editor” section, and submit your "Accept" recommendation.

Reviewer #1: All comments have been addressed

Reviewer #2: All comments have been addressed

2. Is the manuscript technically sound, and do the data support the conclusions?

Reviewer #1: Yes

Reviewer #2: Yes

3. Has the statistical analysis been performed appropriately and rigorously? 

Reviewer #1: Yes

Reviewer #2: Yes

4. Have the authors made all data underlying the findings in their manuscript fully available?

Reviewer #1: Yes

Reviewer #2: Yes

5. Is the manuscript presented in an intelligible fashion and written in standard English?

Reviewer #1: Yes

Reviewer #2: Yes

6. Review Comments to the Author

Reviewer #1: The authors have thoroughly addressed all my comments and suggestions. I am now pleased to recommend this manuscript for acceptance and publication.

Reviewer #2: I commend the authors for thoughtfully addressing my comments. I have no further concerns and recommend acceptance of the manuscript

7. PLOS authors have the option to publish the peer review history of their article (what does this mean? ). If published, this will include your full peer review and any attached files.

**Do you want your identity to be public for this peer review?** For information about this choice, including consent withdrawal, please see our Privacy Policy .

Reviewer #1: No

Reviewer #2: **Yes: ** Dr. Eliala Alice Salvadori

---

## [Editor Report · Acceptance letter]

PONE-D-24-26181R1

PLOS ONE

Dear Dr. Vreden,

I'm pleased to inform you that your manuscript has been deemed suitable for publication in PLOS ONE. Congratulations! Your manuscript is now being handed over to our production team.

Kind regards,

on behalf of

Dr. Milica Nikolic

Academic Editor

PLOS ONE